# ‘Stealth’ Prostate Tumors

**DOI:** 10.3390/cancers15133487

**Published:** 2023-07-04

**Authors:** Vinayak G. Wagaskar, Osama Zaytoun, Swati Bhardwaj, Ash Tewari

**Affiliations:** 1Department of Urology, Icahn School of Medicine at Mount Sinai Hospital, New York, NY 10029, USA; 2Urology Department, Alexandria University, Alexandria 21113, Egypt; 3Department of Pathology, Icahn School of Medicine at Mount Sinai Hospital, New York, NY 10029, USA

**Keywords:** prostate cancer, prostate biopsy, magnetic resonance imaging

## Abstract

**Simple Summary:**

Efforts are ongoing to improve the diagnosis of prostate cancer. Novel blood and tissue-based biomarkers, advanced imaging modalities and image-guided biopsy techniques have further improved cancer detection rates. However, approximately 30–40% of cancers are still missed. Analysis of radical prostatectomy specimens is the only gold standard method for confirming the presence or absence of cancers. In this article, we aim to study those cancers that are missed by standard biopsy techniques and advanced imaging modalities, the so-called ‘Stealth’ prostate cancers. We focus on the lobe of the prostate where cancer is not detected on standard biopsy or by preoperative magnetic resonance imaging (MRI). This article helps to explain the significant false negative rates for current diagnostic modalities for prostate cancer. This will help future research to develop new strategies to improve the detection of these ‘stealth’ tumors.

**Abstract:**

Background: The aim of this study was to determine the false negative rates of prebiopsy magnetic resonance imaging (MRI) and MRI–ultrasound (US) 12-core systematic prostate biopsy (PBx) by analyzing radical prostatectomy specimens. Methods: This retrospective study included 3600 prostate cancer (PCa) patients who underwent robot-assisted laparoscopic radical prostatectomy. Based on comparison of lobe-specific data on final pathology with preoperative biopsy and imaging data, the study population was subdivided into group I—contralateral (CL) benign PBx (*n* = 983), group II—CL and/or bilateral (BL) non-suspicious mpMRI (*n* = 2223) and group III—CL benign PBx + non-suspicious mpMRI (*n* = 688). This population was studied for the presence of PCa, clinically significant PCa (csPCa), extracapsular extension (ECE) (pathological stage pT3), positive frozen section and final positive surgical margin (PSM) in the CL lobe. Descriptive statistics were performed. Results: In subgroups I, II and III, PCa was respectively detected in 21.5%, 37.7% and 19.5% of cases, and csPCa in 11.3%, 16.3% and 10.3% of cases. CL pT3 disease was seen in 4.5%, 4% and 5.5%, and CL surgical margins and/or frozen section analysis were positive in 6%, 7% and 5% of cases in subgroups I, II and III, respectively. Conclusions: There are still significant rates of false negatives in the standard care diagnostics of PCa. Further strategies are required to improve the accuracy of diagnosis and determination of tumor location.

## 1. Introduction

The widespread application of prostate-specific antigen (PSA) screening has led to more cases of prostate cancer (PCa) being diagnosed at an earlier clinical stage [1]. In this scenario, transrectal ultrasound-guided prostate biopsy (TRUS PBx) and subsequent pathological analysis are considered the gold standard for cancer diagnosis. Over the past two decades, the PBx scheme has witnessed numerous modifications to improve the biopsy yield [1,2,3]. Laterally directed extended PBx was found to significantly enhance the diagnosis of PCa compared with conventional sextant biopsy. However, the false negative rate remains substantial [4]. Serefoglou et al. performed repeat 12-core PBx on radical prostatectomy specimens, and surprisingly, the false negative rate of 12-core PBx in this series was 32.2% [5]. This relatively high false negative rate may be intuitively attributed to a limited amount of tissue sampled during PBx, the biopsy surgeon’s experience, the lack of uniform and standardized biopsy techniques and the random nature of biopsy schemes with a resultant sampling error.

Multiparametric magnetic resonance imaging (mpMRI) has emerged as a promising tool for guiding PBx decision-making. The introduction of mpMRI-targeted PBx has increased the accuracy of clinically significant PCa (csPCa) detection [6,7,8]. Nevertheless, The American Urological Association raised concern regarding the risk of missing csPCa on negative mpMRI examinations [9]. We recently published our study on a series of 200 men with negative mpMRI, with 18% found to have PCa and 8% csPCa on the subsequent biopsy [9]. That study was performed with the reference standard as PBx. Our prior work on radical prostatectomy patients comparing suspicious vs. non-suspicious MRI demonstrated similar rates of csPCa, positive surgical margins and biochemical recurrence rates in both groups [10].

Here, we aimed to assess the accuracy of PBx and mpMRI in the diagnosis and localization of PCa and csPCa. This was performed by using lobe-specific final pathological data derived from radical prostatectomy specimens as a reference. We believe this will help to better understand the so-called ‘stealth’ PCa.

## 2. Materials and Methods

### 2.1. Study Design

The study was approved by the Institutional Review Board (GCO#14-0175) of the Icahn School of Medicine at Mount Sinai (New York, NY, USA). We retrospectively reviewed the data of 3600 men who underwent robotic-assessed laparoscopic radical prostatectomy (RALP) between 1 January 2014 and 31 December 2022. RALPs were performed by a single surgeon (A.T.) with more than 20 years of experience in the PCa field.

#### 2.1.1. Inclusion and Exclusion Criteria

Patients who underwent RALP were included in the study if they had full preoperative PBx and mpMRI data. Exclusion criteria encompassed PBx schemes of fewer than 12 systemic cores, contraindications or unreadable mpMRI; prior hormonal or radiation manipulation; preoperative PSA > 20 ng/dL; absence of specific documentation on final pathology or missing information for clinical variables (Figure 1).

#### 2.1.2. MRI Protocol

Prostate evaluations were conducted using 3T MRI Siemens Skyra systems equipped with a phased-array coil. The following sequences were obtained: multiplanar high-resolution T2 fast spin echo (FSE); axial T1 FSE; axial diffusion-weighted imaging; axial T1 in and out of phase; and axial T1 perfusion before and after contrast injection (8 mL of Gadavist (gadobutrol) and 1 mg of glucagon via intramuscular injection). The mpMRI results were evaluated according to Prostate Imaging Reporting and Data System version 2 (PI-RADS v2) [11] by radiologists with more than 5 years of experience in mpMRI prostate imaging (>250 MRI scans per year). Non-suspicious MRI findings were defined as a PI-RADS v2 score of <3.

#### 2.1.3. Biopsy Protocol and Technique

Indications for PBx were one or more of the following: PSA > 4 ng/mL, a 4K score (OPKO Diagnostics, Woburn, MA, USA) of >7%, PSA density of ≥0.15 ng/mL/cm^3^ or suspicious digital rectal examination (DRE).

A transrectal ultrasound was performed. An amount of 5 cc of 1% Lidocaine was injected into each neurovascular bundle (a total of 10 cc was given). Care was taken to avoid intravascular injection. The prostate was then examined and measured using ellipsoid formula. For patients who had mpMRI suspicion (PI-RADS ≥ 3), the Artemis MRI/TRUS fusion device (Innomedicus, Cham, Switzerland) was attached to the ultrasound probe. Range of mobility of the arm was tested to ensure entire prostate from base to apex was reachable for the biopsy. After that, a repeat 360-degree scan was performed of the prostate. Semi-segmentation was performed in both transverse and sagittal views. After this, the MRI was loaded and fused with the ultrasound images.

The target/s on the MRI were identified on ultrasound and the biopsy targets were assigned to that zone. After this, motion artifact and recalibration were corrected under local anesthesia, and four targeted biopsies were taken; path of needle was documented on the Artemis. The targets biopsied were labeled as per location; e.g., Target Right Mid peripheral zone posteromedial. A systemic biopsy was then performed on all 12 quadrants. Systematic biopsies were labeled as Right Lateral Base, Right Medial Base, Right Lateral Mid, Right Medial Mid, Right Lateral Apex, Right Medial Apex, Left Lateral Base, Left Medial Base, Left Lateral Mid, Left Medial Mid, Left Lateral Apex and Left Medial Apex.

All biopsies were performed with a spring-loaded biopsy gun and 18-gauge needles with 12 mm average core length. Cores were placed on non-adherent gauze pad and, finally, in a bottle containing 10% formalin [12,13].

#### 2.1.4. Pathological Assessment

An experienced genitourinary pathologist reviewed both PBx samples and RALP specimens (final pathology). Only H&E slides were reviewed for most cases. PIN-4 staining (including AMACR, high molecular weight cytokeratin and CK5/6) was performed on the few suspicious biopsy cases to confirm the diagnosis of cancer when not sufficiently evident based on morphology alone. The final pathology was comprehensively reviewed on a lobe-specific basis per the College of American Pathologists’ (CAP) protocol for radical prostatectomy specimens [14]. This included comments on the presence of any PCa, Gleason score, grade group, presence of csPCa, presence of extracapsular extension (ECE) (pathological stage pT3), presence of positive neurosafe/frozen section margins and presence of positive surgical margins (PSM).

#### 2.1.5. Outcome Definitions and Statistical Analysis

Gleason grading system was utilized as proposed by Epstein et al. [13], where Grade Group 1  =  Gleason score  ≤  6, Grade Group 2  =  Gleason score 3  +  4  =  7, Grade Group 3  =  Gleason score 4  +  3  =  7, Grade Group 4  =  Gleason score 4  +  4  =  8 and Grade Group 5  =  Gleason scores 9 and 10. PCa is defined as Gleason Grade Group (GGG) 1 and above, while csPCa is defined as GGG ≥ 2 [15]. Based on comparison of lobe-specific data on final pathology with preoperative biopsy and imaging data, study population was further subdivided into three groups (Figure 2):I.Contralateral (CL) benign PBx (*n* = 983).II.CL and/or bilateral (BL) non-suspicious mpMRI (*n* = 2223).III.CL benign PBx + non-suspicious mpMRI (*n* = 688).

This population was studied for presence of any PCa, csPCa, extra-capsular extension (pathological stage pT3), positive frozen section and positive surgical margins in CL lobe (Figure 3).

Descriptive statistics for the three groups were collected. Then, within each group, we compared patients with no CL cancer on final pathology (accurate) versus those with CL cancer (false negative). Results for continuous variables were reported as the median and interquartile range (IQR) and were compared using the Mann–Whitney U test. Results for categorical variables were reported as the frequency and proportion and were compared using a x2 test, as appropriate. All tests were two-tailed with a significance level of *p* < 0.05.

## 3. Results

PCa was detected in 21.5%, 37.7% and 19.5% of the three subgroups, respectively. Detection of csPCa was higher in group II (16.3%) than in the other two groups (11.3% and 10.3%, respectively). Other pathological findings were comparable between the study groups (Table 1).

Table 2 depicts clinical characteristics for group I. Per final pathology, accurate concordance with biopsy results was shown in 78.5% (no CL cancer detected), while CL PCa was diagnosed in the rest, giving a false negative value of 21.5%. Both patient cohorts showed comparable age and median PSA at the time of diagnosis. Of note, the cohort with CL cancer on final pathology had statistically significant higher African American (AA) race, biopsy GGG, pathological T3 stage and PSM.

Regarding group II, accurate and false negative results were encountered in 62.3% and 37.7%, respectively. Patients with CL cancer on final pathology had non-suspicious MRI PI-RADS lesions in 40.2% compared to 1% in patients with no CL cancer cohort (*p* value < 0.001) (Table 3).

False-negative results were encountered in 19.5% of patients in group III. This cohort showed statistically significant pathological T3 stage and PSMs in terms of cohort accurate correlation (Table 4). Figure 4 shows an example of GGG4 cancer that was missed on biopsy and prebiopsy MRI.

## 4. Discussion

Diagnosis of PCa has primarily relied on laboratory tests and MRI followed by MRI-guided PBx. Efforts are ongoing to improve cancer diagnosis by involving biomarkers (4K score, select MDx, etc.) or using nomogram-derived calculators [16,17,18].

Military ‘stealth’ aircraft are designed with fascinating technology that makes their sonar or radar detection challenging. These aircraft are of similar size to other military aircraft but are made of special absorbent materials with unique shapes and contours that cannot be detected by radar [19]. In the current study, we introduce the term ‘stealth PCa’ to describe tumors that are missed on initial evaluation (systematic biopsy, mpMRI) and subsequently diagnosed via a prostatectomy specimen. We utilized lobe-specific final pathology of RALP specimens to assess the actual yield of current standard diagnostics for PCa.

Although the false negative rate of random biopsy protocols is well documented in the literature, we expect that the introduction of MRI and subsequent target biopsy will improve cancer detection and localization. However, counterintuitively, false negative results and, hence, ‘stealth’ tumors were still encountered. We still believe that we are far from precise tumor localization even with extensive expertise in the field of PCa similar to that of our surgeon.

Our study confers three key features: First, a significant number of csPCa cases are missed by mpMRI. It was shown that mpMRI improves the detection of csPCa as well as contributes to reducing the number of unnecessary PBx. Nevertheless, false negative rates of csPCa for non-suspicious MRI range from 2% to 18% [9,20]. We observed a 16% false negative rate in the diagnosis of csPCa in men with non-suspicious MRI, which is in concordance with published studies.

Second, non-suspicious mpMRI and systematic biopsy miss 11–16% of clinically significant ‘stealth’ PCa. The authors believe this is the most critical conclusion of our analysis and should be regarded with extreme caution. We still confirm that mpMRI and subsequent target biopsy have revolutionized the scope of PCa diagnosis. mpMRI provides anatomical and functional details as well as excellent positive predictive and negative predictive values.

In 2017, the PROMIS trial demonstrated that using mpMRI as an initial triage for men with an elevated PSA could allow 27% of patients to avoid a primary biopsy and diagnose 5% fewer clinically insignificant cancers. When compared to using a standard TRUS-guided PBx pathway, using mpMRI to guide biopsy can allow urologists to detect 18% more cases of significant cancers [21]. Hence, an MRI-guided biopsy followed by a systematic biopsy has been the gold standard in PCa diagnosis. Therefore, it is still an integral part of our own current practice that involves routine prebiopsy mpMRI. We believe that larger prospective studies are still needed to validate this critical conclusion.

Sampling error on needle PBx has been well demonstrated in the literature. It is due to a small amount of tissue (approximately 0.04% of the average gland volume) that is removed by thin-core needle biopsies [22]. Therefore, false negative rates are commonly encountered. Some authors confirmed a statement similar to our second conclusion. Kim et al. studied 730 radical prostatectomy specimens and compared them to combined systemic TRUS PBx of at least 12 cores and mpMRI. They concluded that this combination did not provide reliable accuracy in predicting the true unilaterality of PCa [23]. In another study by the same group, the sensitivity, specificity, positive predictive value and negative predictive value of mpMRI to predict csPCa were 74.3%, 45.5%, 95.5% and 10.2%, respectively [24].

We hypothesize that these challenges in accurate preoperative PCa localization may be attributed to the high heterogeneity and multifocality behavior of the disease itself. Additionally, this confirms that even with these marvelous advances in PCa diagnostics, we still lack the best tools for precise cancer detection and localization. We have initiated a trial on the wide application of micro-ultrasound to better localize PCa preoperatively; however, details of such trials are beyond the scope of this study.

Third, the presence of AA race and biopsy GGG ≥ 2 increases the possibilities of CL ‘stealth’ tumors. Molecular and genomic differences in the tumor biology of AA men have been widely studied to explain the aggressiveness and increased incidences of PCa compared with the non-Hispanic white population. Our prior work on AA men showed increased biopsy GGG upgrading and increased incidences of biochemical recurrence compared with other men. Herein, we found that AA men have increased incidences of ‘stealth’ tumors. The Gleason grading system is still a commanding predictor of PCa; the higher the grade, the worse the outcome. In our series, patients with GGG ≥ 2 have an increased incidence of CL stealth tumors. Surprisingly, our study also highlighted that there is no significance due to age, median PSA, family history of PCa or median prostate volume in determining the presence of CL ‘stealth’ tumors; i.e., we found no correlation of age, median PSA or prostate volume in men with vs. without CL ‘stealth’ tumors.

This study has its limitations. Firstly, it was retrospective in nature, where all the data were derived from our database. Second, the preoperative MRI/TRUS-guided 12-core PBxs were not performed by the same physician. Although there is no evidence to support any differences in the results of PBx between the performing urologists, we believe this factor may have influenced our data. Lastly, this study included only men with previously positive PBx that were recommended for and then underwent RALP and, therefore, excluded men with false negative initial biopsy, clinically insignificant prostate cancers not requiring RALP and others that underwent different treatment options (radiation, focal therapy, hormonal therapy, etc.). Therefore, the actual risk of a false negative biopsy may be much higher, and further studies are required to address this confounding factor.

## 5. Conclusions

The current standards of care for diagnostics for PCa (PSA, DRE, MRI and MRI–US-guided prostate biopsy or 12-core systematic biopsy) have significant false negative rates. Further strategies are required to improve the accuracy of diagnosis.

## Figures and Tables

**Figure 1 cancers-15-03487-f001:**
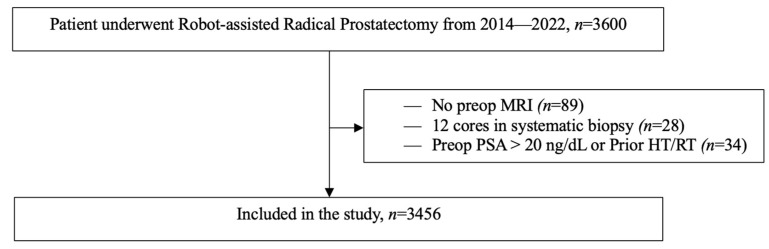
Flow chart depicting inclusion and exclusion criteria used to derive patient population used for data analysis. Abbreviations: MRI: magnetic resonance imaging, PSA: prostate-specific antigen, HT: Hormone therapy, RT: Radiation therapy.

**Figure 2 cancers-15-03487-f002:**
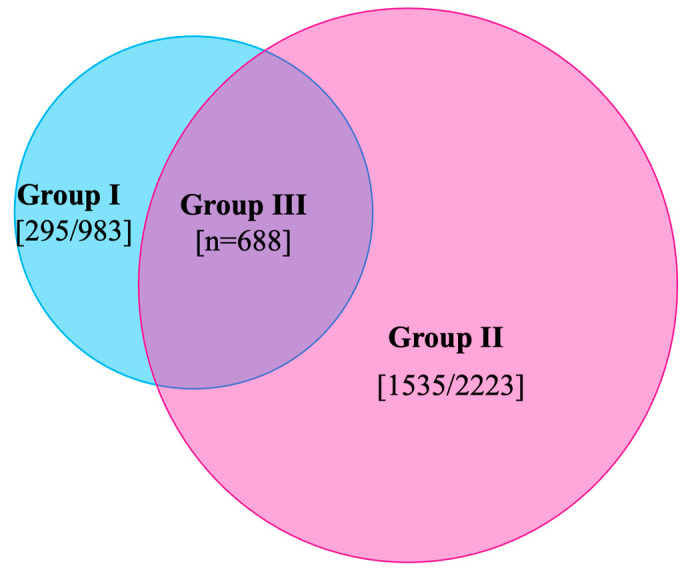
Venn diagram showing distribution of study population in each group. Group III population is an overlap between group I and group II.

**Figure 3 cancers-15-03487-f003:**
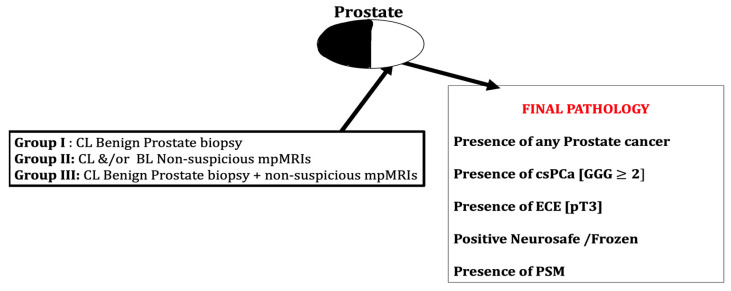
Schematic representation of methodology and study population classification. Patients with contralateral lobe benign prostate biopsy and/or non-suspicious MRI were studied for presence of cancer and other variables in radical prostatectomy specimens. Abbreviations: CL—contralateral, BL—bilateral, mpMRI—multiparametric magnetic resonance imaging, csPCa—clinically significant prostate cancer, ECE—extracapsular extension of cancer, PSM—positive surgical margins.

**Figure 4 cancers-15-03487-f004:**
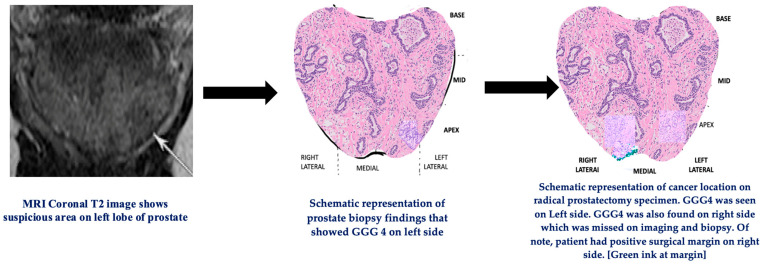
Prebiopsy MRI followed by schematic representation of GGG4 cancer prostate biopsy and radical prostatectomy. Patient had cancer on right side that was missed on prebiopsy MRI and prostate biopsy. Abbreviations: MRI—magnetic resonance imaging, GGG—Gleason Grade Group.

**Table 1 cancers-15-03487-t001:** Comparative analysis between the three subgroups regarding the final pathological findings.

Final Pathology Parameters	Group I: *n* = 983 (%)	Group II: *n* = 2223 (%)	Group III: *n* = 688 (%)
Presence of any PCa, no (%)	212 (21.5)	825 (37.7)	134 (19.5)
Presence of csPCa, no (%)	111 (11.3)	362 (16.3)	71 (10.3)
Presence of ECE (pT3), no (%)	45 (4.5)	85 (3.8)	38 (5.5)
Positive frozen section analysis, no (%)	51 (5.2)	102 (4.5)	27 (4)
Presence of PSMs, no (%)	18 (1.8)	61 (2.7)	11 (1.5)

Abbreviations: PCa—prostate cancer, csPCa—clinically significant prostate cancer, ECE—extracapsular extension of cancer, PSM—positive surgical margins.

**Table 2 cancers-15-03487-t002:** Comparison between cohorts with accurate versus false negative results in group I.

Variable	Patients with No CL Cancer on Final Pathology: Accurate, *n* = 771 (78.5%)	Patient with CL Cancer on Final Pathology: False Negative, *n* = 212 (21.5%)	*p* Value
Median age in years	64	63	0.429
Race			<0.023 *
AA	80 (10.4)	37 (17.5)	
White	437 (56.7)	119 (56.1)	
Others	254 (32.9)	56 (26.4)	
BMI	26.9	27.1	0.362
Family history of PCa			0.320
No	582 (75.5)	164 (77.4)	
Yes	189 (24.5)	48 (22.6)	
Median PSA at diagnosis	6.1	6.3	0.258
Median prostate volume (cc)	40	39	0.381
Biopsy GGG			0.019 *
1	101 (13.1)	21 (9.9)	
2	295 (38.3)	93 (43.9)	
3	201 (26.1)	51 (24.1)	
4	122 (15.8)	22 (10.4)	
5	52 (6.7)	25 (11.8)	
MRI PI-RADS lesions			0.129
1–2	105 (13.6)	23 (10.8)	
3	81 (10.5)	13 (6.1)	
4	368 (47.7)	107 (50.5)	
5	217 (28.1)	69 (32.5)	
Final pathology GGG			0.056
1	71 (9.2)	16 (7.5)	
2	412 (53.4)	103 (48.6)	
3	202 (26.2)	63 (29.7)	
4	38 (4.9)	10 (4.7)	
5	48 (6.2)	20 (9.4)	
Pathology T stage			<0.001 *
T2	660 (85.6)	133 (62.7)	
T3	111 (14.4)	79 (37.3)	
PSMs			<0.001 *
Absent	736 (95.5)	187 (88.2)	
Present	35 (4.5)	25 (11.8)	

Abbreviations: CL—contralateral, AA—African American race, PCa—prostate cancer, PSA—prostate specific antigen, GGG—Gleason Grade Group, MRI—magnetic resonance imaging, PI-RADS—Prostate Imaging Radiology And Data System, PSM—positive surgical margins. * *p* value < 0.05.

**Table 3 cancers-15-03487-t003:** Comparison between cohorts with accurate versus false negative results in group II.

Variable	Patients with No CL Cancer on Final Pathology: Accurate *n* = 1398 (62.3%)	Patient with CL Cancer on Final Pathology: False Negative *n* = 825 (37.7%)	*p* Value
Median age in years	64	63	0.429
Race			<0.001 *
AA	158 (11.3)	131 (15.9)	
White	855 (61.2)	497 (60.2)	
Others	385 (22.4)	197 (23.9)	
BMI	26.9	27.1	0.362
Family history of PCa			0.067
No	1072 (76.7)	656 (79.5)	
Yes	326 (23.3)	169 (20.5)	
Median PSA at diagnosis	6.0	6.0	0.258
Median prostate volume (cc)	39	39	0.381
Biopsy GGG			0.032 *
1	225 (16.1)	175 (21.2)	
2	553 (39.6)	323 (39.2)	
3	305 (21.8)	165 (20.0)	
4	196 (14.0)	98 (11.9)	
5	119 (8.5)	64 (7.8)	
MRI PI-RADS lesions			<0.001 *
1–2	13(1)	332 (40.2)	
3	162 (11.6)	62 (7.5)	
4	716 (51.2)	252 (30.5)	
5	507 (36.3)	179 (21.7)	
Final pathology GGG			0.017 *
1	127 (9.1)	98 (11.9)	
2	764 (54.6)	480 (58.2)	
3	344 (24.6)	174 (21.1)	
4	60 (4.3)	24 (2.9)	
5	103 (7.4)	49 (5.9)	
Pathology T stage			<0.001 *
T2	1154 (82.5)	570 (69.1)	
T3	244 (17.5)	255 (30.9)	
PSMs			<0.001 *
Absent	1336 (95.6)	738 (89.5)	
Present	62 (4.4)	87 (10.5)	

Abbreviations: CL—contralateral, AA—African American race, PCa—prostate cancer, PSA—prostate-specific antigen, GGG—Gleason Grade Group, MRI—magnetic resonance imaging, PI-RADS—Prostate Imaging Radiology And Data System, PSM—positive surgical margins. * *p* value < 0.05.

**Table 4 cancers-15-03487-t004:** Comparison between cohorts with accurate versus false negative results in group III.

Variable	Patients with No CL Cancer on Final Pathology: Accurate, *n* = 554 (81.5%)	Patient with CL Cancer on Final Pathology: False Negative, *n* = 134 (19.5%)	*p* Value
Median age in years	64	63	0.429
Race			0.093
AA	53 (9.6)	19 (14.2)	
White	314 (56.7)	82 (61.2)	
Others	187 (33.8)	33 (24.6)	
BMI	26.9	27.1	0.362
Family history of PCa			0.320
No	416 (75.1)	103 (76.9)	
Yes	138 (24.9)	31 (23.1)	
Median PSA at diagnosis	6.1	6.3	0.258
Median prostate volume (cc)	40	39	0.381
Biopsy GGG			0.001 *
1	66 (11.9)	4 (3)	
2	210 (37.9)	57 (42.5)	
3	142 (25.6)	44 (32.8)	
4	92 (16.6)	13 (9.7)	
5	44 (7.9)	16 (11.9)	
MRI PI-RADS lesions			0.129
1–2	5 (1)	2 (1)	
3	57 (10.3)	8 (6)	
4	317 (57.2)	72 (53.7)	
5	175 (31.6)	52 (38.8)	
Final pathology GGG			0.056
1	71 (9.2)	16 (7.5)	
2	412 (53.4)	103 (48.6)	
3	202 (26.2)	63 (29.7)	
4	38 (4.9)	10 (4.7)	
5	48 (6.2)	20 (9.4)	
Pathology T stage			<0.001 *
T2	465 (83.9)	81(60.4)	
T3	89 (16.1)	53 (39.6)	
PSMs			<0.001 *
Absent	532 (96)	118 (88.1)	
Present	22 (4)	16 (11.9)	

Abbreviations: CL—contralateral, AA—African American race, PCa—prostate cancer, PSA—prostate specific antigen, GGG—Gleason Grade Group, MRI—magnetic resonance imaging, PI-RADS—Prostate Imaging Radiology And Data System, PSM—positive surgical margins. * *p* value < 0.05.

## Data Availability

The data presented in this study are available on request from the corresponding author. The data are not publicly available due to ethical restrictions.

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
