# Peer review of "‘Stealth’ Prostate Tumors"

_cancers, 2023, doi:10.3390/cancers15133487_

Round 1
Reviewer 1 Report
The paper is of great interest due to identifying and characterizing MR and biopsy invisible lesions. I have the following comments and I hope the authors consider improving the presentation based on those:
1- Graphical abstract and Figure 3, both are very confusing. The outline of prostate gland in MR and the histopathology seems to be not matching. Furthermore, histopathology image scale seems to be not quite right. I suggest for authors to make sure the specimen outline and gland outline match, and also in Figure 3, outline of lesion on contralateral seems to be missing.
2- I am also missing the breakdown on the "stealth" lesions grades as compared to detected lesion on contralateral side. It is extremely important to know whether the "stealth" lesion are generally of lower grade or not. The discussion around extending the findings to biopsy-naive patients will be interesting. If that is true, the statement of mp-MRI or biopsy missing significant number of csPCA need to be revised.
Author Response
The paper is of great interest due to identifying and characterizing MR and biopsy invisible lesions. I have the following comments and I hope the authors consider improving the presentation based on those:
- Graphical abstract and Figure 3, both are very confusing. The outline of prostate gland in MR and the histopathology seems to be not matching. Furthermore, histopathology image scale seems to be not quite right. I suggest for authors to make sure the specimen outline and gland outline match, and also in Figure 3, outline of lesion on contralateral seems to be missing.
Reply: Thank you for your suggestions. We now have corrected Axial MR images to T2 Coronal image and matched outlines of MR and histopathology. Also we made correction in Figure 3.
- I am also missing the breakdown on the "stealth" lesions grades as compared to detected lesion on contralateral side. It is extremely important to know whether the "stealth" lesion are generally of lower grade or not. The discussion around extending the findings to biopsy-naive patients will be interesting. If that is true, the statement of mp-MRI or biopsy missing significant number of csPCA need to be revised.
Reply: Thank you for your suggestions. As per your suggestions, Stealth lesion grades are mentioned in Table 1. Yes, stealth lesion grades are usually low grade. In subgroups I [benign biopsy], II [non-suspicious MRI] and III [combined benign biopsy and non-suspicious MRI], csPCa was seen in 11.3%, 16.3% and 10.3% of CL lobes, respectively. I completely agree regarding decision to extend findings to biopsy naïve patients but we do not have biopsy naïve cohort for identifying stealth tumors.

Reviewer 2 Report
This paper shows the false negative of CL cancers missed by TRUS biopsy and/or MRI, there are two main comments for this paper:
1: this paper only presented the CL cancers, which means the whole prostate is only divided into two parts (left and right). Prostate is a complicated organ and according to different diagnostic procedures, it can be divided in to central/peripheral/transitional, lateral/medial, posterior/anterior, and apex/mid/base. It will be better to mention any of methods above rather than only contralateral cancers, For example, if cancer was found in left apex on final pathology but biopsy and/or MRI only detected left bottom lesion, this will be counted as no CL cancer according to the method of this paper but apparently the left apex cancer is still a ‘stealth’ cancer.
2: Biopsy methods are not clear, in line 104-105, systematic biopsy and fusion biopsy are both mentioned but there is no data to show the distribution (how many had systematic biopsy and how many had fusion biopsy, maybe there are some had both?), fusion biopsy method is missing either, how many MRI findings are targeted? how many biopsy cores were taken from each cores, what technologies were used? As we all know, systematic biopsy with/without targeted biopsy will affect the results significantly. I suggest authors dividing them into different groups for separate analyses.
The best method to analyse ‘stealth’ cancer is to locate each lesion from biopsy and MRI and then compare with final pathology results, Authors have shown us detailed lesion maps both from biopsy and final pathology in Fig 3, which means it is feasible to locate at least the primary lesions (anterior/posterior, apex/base), left/right or just CL or not is not enough to support the conclusion.
Comments in lines:
Line 42: the graphical abstract, how the schematic representation (or the pathology map) was made according to 12 biopsy cores or targeted biopsies? More descriptions are needed here.
Line 77-78: author mentioned a single surgeon did 3600 LRP in 9 years, which is 400 per year. Could I know how long each surgery takes in your hospital?
Line 91: MRI brand is missing here.
Line 98: internal or external validation needs to be done to verify the MRI data.
Line 105: biopsy methods are not clearly described. See main comment 2.
Line 109: the same pathologist reviewed both biopsy and final pathology results which will cause biased results. The biopsy results, MRI results and final pathology results need to be double blinded and then collected and analysed by a third party.
Line 120, Gleason Grade Group is mentioned here but there it no reference and definition to show what it is, same for csPAs.
Line 123-126: the numbers don’t match, which may invalid all analyses.
Line 185-186, figure 3, the second and third pathology map are similar? Any description to show how this map was drawn based on 12 biopsy cores?
Author Response
This paper shows the false negative of CL cancers missed by TRUS biopsy and/or MRI, there are two main comments for this paper:
1: this paper only presented the CL cancers, which means the whole prostate is only divided into two parts (left and right). Prostate is a complicated organ and according to different diagnostic procedures, it can be divided in to central/peripheral/transitional, lateral/medial, posterior/anterior, and apex/mid/base. It will be better to mention any of methods above rather than only contralateral cancers, For example, if cancer was found in left apex on final pathology but biopsy and/or MRI only detected left bottom lesion, this will be counted as no CL cancer according to the method of this paper but apparently the left apex cancer is still a ‘stealth’ cancer.
Reply: Thank you for your suggestions. We agree with the reviewer regarding the complexity of the zonal anatomy of prostate gland. However, we still opted to regard it as divided into right and left lobes. This is simply because the TRUS biopsy templates have been always using the same description. Urologists have been trained since the residency to obtain right lobe cores and left lobe cores. This is the settled practice up till now. This templates certainly include base, mid and apical cores. We opted not to include these sub-classification to keep the simplicity of our analysis and avoid any further confusion. However, the reviewer’s comment is valuable and can be applied for further future studies with more sophisticated analyses
2: Biopsy methods are not clear, in line 104-105, systematic biopsy and fusion biopsy are both mentioned but there is no data to show the distribution (how many had systematic biopsy and how many had fusion biopsy, maybe there are some had both?), fusion biopsy method is missing either, how many MRI findings are targeted? How many biopsy cores were taken from each cores, what technologies were used? As we all know, systematic biopsy with/without targeted biopsy will affect the results significantly. I suggest authors dividing them into different groups for separate analyses.
Reply: Thank you for the reviewer comment. Regarding the Systemic biopsy, we follow the 12 –core extended biopsy scheme including base, mid and apical cores on each side (Right and left Lobes). For each zone we obtain two cores (medial and lateral). All 12 cores were kept separately for analysis. Regarding the target biopsy and in additional to the previously mentioned systematic scheme, we obtain 4 cores for each PIRADS ≥ 3 lesion. The 4 cores for each individual lesion were kept together in the same specimen for pathological analysis assigned to this specific lesion. Regarding the distribution of biopsies between systematic vs. target, any patient with PIRADS ≥ 3 lesion has combined target and systematic biopsies. Table 2, 3 and 4 depicts PIRADS lesions breakdown in respective patient groups. Those patients with suspicious lesions on MP-MRI underwent MRI/US fusion-guided targeted biopsy using the Artemis MRI/TRUS fusion device (Innomedicus, Cham, Switzerland). This now has been mentioned in the manuscript as per suggestion.
- The best method to analyse ‘stealth’ cancer is to locate each lesion from biopsy and MRI and then compare with final pathology results, Authors have shown us detailed lesion maps both from biopsy and final pathology in Fig 3, which means it is feasible to locate at least the primary lesions (anterior/posterior, apex/base), left/right or just CL or not is not enough to support the conclusion.
Reply: The aim of the study and as mentioned in the end of introduction section, to assess the accuracy of PBx and mpMRI in diagnosis and localization of PCa and csPCa. This was performed by using lobe-specific final pathological data derived from radical prostatectomy specimens as a reference. This was the hypothesis that was set by the authors and hence the study design was tailored to achieve this target. We appreciate the authors’ comment and we believe it will be considered in future studies with different study design to start with.
Comments in lines:
Line 42: the graphical abstract, how the schematic representation (or the pathology map) was made according to 12 biopsy cores or targeted biopsies? More descriptions are needed here.
Reply: The graphical abstract shows a case in which mp-MRI depicted high suspicious lesion, so the schematic representation was made according to systematic 12 core PBx plus target biopsy cores.
Line 77-78: author mentioned a single surgeon did 3600 LRP in 9 years, which is 400 per year. Could I know how long each surgery takes in your hospital?
Reply: The current study demonstrates a single surgeon experience (A.T). The surgeon is considered a figure in RALP over the country, being in a tertiary referral centre made his volume of 400-500 cases a year. The majority of cases are discharged at the morning next to surgery day.
Line 91: MRI brand is missing here.
Reply: 3‐T MRI Siemens Skyra equipped with a phased‐ array coil was utilized. This now has been mentioned in the manuscript.
Line 98: internal or external validation needs to be done to verify the MRI data.
Reply: while this is beyond the scope of the current study, we previously published our series to verify MRI data in another separate study specifically designated for this purpose. Please review reference “9”.
Line 105: biopsy methods are not clearly described. See main comment 2.
Reply: Thank you for your suggestions. this was addressed in comment 2
Line 109: the same pathologist reviewed both biopsy and final pathology results which will cause biased results. The biopsy results, MRI results and final pathology results need to be double blinded and then collected and analysed by a third party.
Reply: As per our institutional protocol, the prostate biopsy and prostatectomy specimens are assigned to a specific pathologist with an extensive experience in the field of Prostate cancer. We disagree with the reviewer and we believe that assigning biopsy and final pathology to different pathologists will be intuitively impacted by unavoidable difference in experience and will ultimately lead to bias and heterogeneous results. To support this opinion, we previously published multiple studies that we included our same pathologist in first class urology journals and we did not receive similar comments “please see reference 9” as an example.
Line 120, Gleason Grade Group is mentioned here but there it no reference and definition to show what it is, same for csPAs.
Reply: Thank you for your suggestions. This now has been mentioned in detail with the reference
Line 123-126: the numbers don’t match, which may invalid all analyses.
Reply: We reviewed the number and it matches our study cohorts. Of note, these numbers are lobe specific.
Line 185-186, figure 3, the second and third pathology map are similar? Any description to show how this map was drawn based on 12 biopsy cores?
Reply: the figure was reviewed, and second and third pathology map are totally different as mentioned in the description below each map. Second map shows prostate biopsy finding of GG4 on left side. While third map shows final pathology on radical prostatectomy specimen of the same patient showing GG4 on left side as well as GG4 on right side and additionally positive surgical margin on right side. This scenario clearly demonstrates simplifies our main of the study to better understand “Stealth “contralateral cancers.

Reviewer 3 Report
Comments;
Clinically useful study. Large case series. Text and Tables are clear. Illustrations are mostly reasonable quality.
1. Graphic Abstract: Image 2, Axial MRI – saying suspicious area in the left prostate. Wrong, this is a normal prostate MRI imaging. Also, all other images are in coronal plane, so the author should ask a radiologist to help finding the same patient’s MRI, T2 coronal view with suspicious area showing here.
2. In practice, there are many patients who have more than one lesions in bilateral lobes who underwent MRI and then biopsies with positive results. This should be the largest group of patients who underwent prostatectomy. I did not see authors include these patients in their inclusion or exclusion criteria, or I just did not understand the author’s study group II population. Please explain.
3. Line 97 “….by a radiologist with more than 5 years of experience”. In a hospital with 3600 prostatectomies in 8 years, only one radiologist read all prostate MRI is very difficult to believe. When the radiologist went to vacation, who read them?
4. Figure 3, Image 1 – Line 185, axial t2 MRI. Other 2 images are in coronal views. It should replace with a coronal view T2 MRI with lesion.
5. “We observed 16% false negative rate in diagnosis of csPCa in men with non-suspicious MRI which is in concordance published series.” Line 208. That is not too high, particularly considering the fact that when a patient had a dominant PCa on one lobe, the less aggressive PCa on the other lobe might be not obvious at MRI due to suppression of functional imaging by the dominant tumor. These missed cases by MRI possibly did not influence the patients’ treatment and outcome since they were not the dominant tumors in these patients.
6. Line 210 “Second, mpMRI adds minimal or no benefit to standard 12-core biopsy in diagnosing 210 ‘stealth’ PCa.” In my opinion, it is a wrong conclusion. This study did not have power to draw this conclusion. In our experience and research, standard 12-core biopsy might miss up to 25% to 35% clinically significant PCa; majority of them in the anterior aspect or in the apex. In comparison, MRI might miss up to 10% clinically significant PCa (bases on our hospital data). So the advantage of prostate MRI is obvious.
Author Response
Clinically useful study. Large case series. Text and Tables are clear. Illustrations are mostly reasonable quality.
Graphic Abstract: Image 2, Axial MRI – saying suspicious area in the left prostate. Wrong, this is a normal prostate MRI imaging. Also, all other images are in coronal plane, so the author should ask a radiologist to help finding the same patient’s MRI, T2 coronal view with suspicious area showing here.
Reply: Thank you for your suggestions. We now have replaced Axial image with T2 Coronal images both in Graphical Abstract and Figure 3.
- In practice, there are many patients who have more than one lesions in bilateral lobes who underwent MRI and then biopsies with positive results. This should be the largest group of patients who underwent prostatectomy. I did not see authors include these patients in their inclusion or exclusion criteria, or I just did not understand the author’s study group II population. Please explain.
Reply: Group II encompasses Contralateral and/or bilateral non-suspicious mpMRI (n=2223). Regarding the group of patients pointed at by the reviewer (who have more than one lesion in bilateral lobes), if bilateral lesions are non-suspicious or have a low suspicious on one side and the other side has high suspicious lesion, there were still included in our group II cohort. In case they have bilateral highly suspicious lesions, those were not included in the study.
- Line 97 “….by a radiologist with more than 5 years of experience”. In a hospital with 3600 prostatectomies in 8 years, only one radiologist read all prostate MRI is very difficult to believe. When the radiologist went to vacation, who read them?
Reply; Thank you for your suggestions. We have corrected the statement. It is true that these radiologists are experienced in MR reading for more than 5 years. To support this opinion, we previously published multiple studies that we included our same radiologist in first class urology journals and we did not receive similar comments “please see reference 9” as an example.
- Figure 3, Image 1 – Line 185, axial t2 MRI. Other 2 images are in coronal views. It should replace with a coronal view T2 MRI with lesion.
Reply: Thank you for your suggestions. We now have replaced Axial image with T2 Coronal images both in Graphical Abstract and Figure 3.
- “We observed 16% false negative rate in diagnosis of csPCa in men with non-suspicious MRI which is in concordance published series.” Line 208. That is not too high, particularly considering the fact that when a patient had a dominant PCa on one lobe, the less aggressive PCa on the other lobe might be not obvious at MRI due to suppression of functional imaging by the dominant tumor. These missed cases by MRI possibly did not influence the patients’ treatment and outcome since they were not the dominant tumors in these patients.
Reply: we agree with the reviewer. In this paragraph we did not mention it is considered a “too high” percentage, we just clarified the false negative rate encountered in our cohort and pointed it is still corresponding to the published data in the literature too.
- Line 210 “Second, mpMRI adds minimal or no benefit to standard 12-core biopsy in diagnosing 210 ‘stealth’ PCa.” In my opinion, it is a wrong conclusion. This study did not have power to draw this conclusion. In our experience and research, standard 12-core biopsy might miss up to 25% to 35% clinically significant PCa; majority of them in the anterior aspect or in the apex. In comparison, MRI might miss up to 10% clinically significant PCa (bases on our hospital data). So the advantage of prostate MRI is obvious.
Reply: While we acknowledge this data stated by the reviewer, however our conclusions were derived from our own data and statistical analysis. Worthy to mention, the concept, sample size and statistical methods should be regarded when comparing the current study to other studies in the literature. Hence the ultimate conclusions should be taken with extreme caution respecting each study design. Nevertheless, our false negative rate in diagnosis of csPCa in men with non-suspicious MRI remained in concordance published series. We have modified the statement to minimal benefit and removed no benefit from the draft.

Round 2
Reviewer 1 Report
The information as to whether or not the "stealth lesions" are PC, csPC, or margin positive, as compared to index lesion found as part of mp-MR and targeted and systematic biopsy. This is at core of clinical impact and I encourage the authors to provide that information.
Again, the graphical abstract is poorly done, the image resolution does not allow proper correspondence to be made between the pathology and MR findings. Please kindly improve the presentation.
Author Response
Reviewer 1
The information as to whether or not the "stealth lesions" are PC, csPC, or margin positive, as compared to indexlesion found as part of mp-MR and targeted and systematic biopsy. This is at core of clinical impact and I encourage the authors to provide that information.
Reply: Thank you for suggestion. This suggestion has provided important key feature for this paper. Our aim was only to focus on lobes that were biopsy benign and / or non-suspicious MRI. We went back and looked at dataset to compare the findings to Index lesions found as a part of MRI-US Targeted and systematic biopsy. Our groups I and III matches with the observation as Group II has some men with non-suspicious MRI [whole prostate] that cannot be targeted biopsied. Our rates of findings PCa in MRI and Benign non-suspicious areas are 19.5-21.5%, Finding csPCa are 10.3-11.3%, positive margins 4.5-5.5%. All MRI targeted areas [Index] on the CL side Found PCa on biopsy as all these patients undergone RALP procedure. Stealth is concept where lesions not seen on MRI or not identified on standard systematic biopsy cores.
Again, the graphical abstract is poorly done, the image resolution does not allow proper correspondence to be made between the pathology and MR findings. Please kindly improve the presentation.
Reply: Thank you for suggestion. Graphical abstract resolution is modified as per suggestion.

Author Response
Reviewer 2
This paper shows the false negative of CL cancers missed by TRUS biopsy and/or MRI, there are two main comments for this paper:
1: this paper only presented the CL cancers, which means the whole prostate is only divided into two parts (left and right). Prostate is a complicated organ and according to different diagnostic procedures, it can be divided in to central/peripheral/transitional, lateral/medial, posterior/anterior, and apex/mid/base. It will be better to mention any of methods above rather than only contralateral cancers, For example, if cancer was found in left apex on final pathology but biopsy and/or MRI only detected left bottom lesion, this will be counted as no CL cancer according to the method of this paper but apparently the left apex cancer is still a ‘stealth’ cancer.
Reply: Thank you for your suggestions. We agree with the reviewer regarding the complexity of the zonal anatomy of prostate gland. However, we still opted to regard it as divided into right and left lobes. This is simply because the TRUS biopsy templates have been always using the same description. Urologists have been trained since the residency to obtain right lobe cores and left lobe cores. This is the settled practice up till now. This templates certainly include base, mid and apical cores. We opted not to include these sub-classification to keep the simplicity of our analysis and avoid any further confusion. However, the reviewer’s comment is valuable and can be applied for further future studies with more sophisticated analyses
Over-simplification of the scenario will not clarify the problem which will indeed cause confusion to other researchers, but I agree with authors that it will be easy to explain to patients. What I mean in my previous review is that we could get a simple conclusion but we could not simplify our methods. “The aim or this study was to determine the false negative rates of prebiopsy magnetic resonance imaging (MRI) and MRI–ultrasound (US) 12-core systematic prostate biopsy 23 (PBx) by analyzing radical prostatectomy specimens”, this is a very ambitious aim but your methods couldn’t strongly support this aim.
Reply: Thank you for suggestion. We agree our methodology description was simplified. Thank you for providing reference paper https://onlinelibrary.wiley.com/doi/10.1111/iju.13722
We now have described detailed biopsy method, assessment and pathological diagnosis at our institution. We have also cited the above reference for methodology template. Also, we have mentioned in the limitation as this is selective cohort included only men with previously positive PBx that were recommended for and then underwent RALP and therefore excluded men with false negative initial biopsy, clinically insignificant prostate cancers not requiring RALP and others that underwent different treatment options (radiation, focal therapy, hormonal therapy, etc.).
2: Biopsy methods are not clear, in line 104-105, systematic biopsy and fusion biopsy are both mentioned but there is no data to show the distribution (how many had systematic biopsy and how many had fusion biopsy, maybe there are some had both?), fusion biopsy method is missing either, how many MRI findings are targeted? How many biopsy cores were taken from each cores, what technologies were used? As we all know, systematic biopsy with/without targeted biopsy will affect the results significantly. I suggest authors dividing them into different groups for separate analyses.
Reply: Thank you for the reviewer comment. Regarding the Systemic biopsy, we follow the 12 –core extended biopsy scheme including base, mid and apical cores on each side (Right and left Lobes). For each zone we obtain two cores (medial and lateral). All 12 cores were kept separately for analysis. Regarding the target biopsy and in additional to the previously mentioned systematic scheme, we obtain 4 cores for each PIRADS ≥ 3 lesion. The 4 cores for each individual lesion were kept together in the same specimen for pathological analysis assigned to this specific lesion. Regarding the distribution of biopsies between systematic vs. target, any patient with PIRADS ≥ 3 lesion has combined target and systematic biopsies. Table 2, 3 and 4 depicts PIRADS lesions breakdown in respective patient groups. Those patients with suspicious lesions on MP-MRI underwent MRI/US fusion-guided targeted biopsy using the Artemis MRI/TRUS fusion device (Innomedicus, Cham, Switzerland). This now has been mentioned in the manuscript as per suggestion.
Thanks for your explanation. The reason I asked this question is because the figures in tables are not matched, that’s why I hope authors could review their results carefully. For example, Group 3 are patients with non-suspicious mpMRI which means all patients’ PIRADS <3 but table 4 shows most of patients’ PIRADS >=3.
I’ve carefully reviewed all your 3 tables (table 2 -4). All 3456 eligible patients had prebiopsy MRI and those who were MRI positive (PIRADS>=3) had both systematic biopsy and targeted biopsy, MRI negative patients only had systematic biopsy. All patients’ biopsy were positive because they had LRP. Group 1 (n=983) are CL benign biopsy patients, which mean the rest of 2473 patients (n=3456-983) are CL positive biopsy patients; Group 2 (n=2333) is CL and/or bilateral (BL) non-suspicious mpMRI, does it mean this group of patients is MRI negative or they only have one lesion on left/right side? I suggest authors to expand the flow chart and add all information then I could know how those 3456 patients are distributed, then I will know how these 3 groups are picked up from the flow chart. Then we will know how many patients have systematic biopsy only and how many had both systematic and targeted biopsy. They are very important information and couldn’t be ignored. Similar like below.
Reply : Thank you for your comments. All 3456 eligible patients had prebiopsy MRI and those with suspicious MRI underwent MRI-US Fusion prostate biopsy. The cohort is Radical prostatectomy, hence all of them were having prostate cancer diagnosis. Three Groups that were mentioned in the methodology have overlapping population. Also Group II included patients with either No MRI lesion in the Prostate or unilateral MRI lesion in the prostate so you will see PI-RADS score in this Group as well. It means that 3456 – 2333= 1123 men had lesions on both sides of prostate. To demonstrate the overlap, we can use Venn diagram as seen below. [This also has been mentioned in the draft]. Group III is overlap between Groups I & II.
- The best method to analyse ‘stealth’ cancer is to locate each lesion from biopsy and MRI and then compare with final pathology results, Authors have shown us detailed lesion maps both from biopsy and final pathology in Fig 3, which means it is feasible to locate at least the primary lesions (anterior/posterior, apex/base), left/right or just CL or not is not enough to support the conclusion.
Reply: The aim of the study and as mentioned in the end of introduction section, to assess the accuracy of PBx and mpMRI in diagnosis and localization of PCa and csPCa. This was performed by using lobe-specific final pathological data derived from radical prostatectomy specimens as a reference. This was the hypothesis that was set by the authors and hence the study design was tailored to achieve this target. We appreciate the authors’ comment and we believe it will be considered in future studies with different study design to start with.
Comments in lines:
Line 42: the graphical abstract, how the schematic representation (or the pathology map) was made according to 12 biopsy cores or targeted biopsies? More descriptions are needed here.
Reply: The graphical abstract shows a case in which mp-MRI depicted high suspicious lesion, so the schematic representation was made according to systematic 12 core PBx plus target biopsy cores.
Please refer to the last comment.
Reply: The graphical abstract does not show exact image from each core. It is meant to be a schematic/representative depiction. For instance, if out of 12 biopsy cores, 11 were benign, and one, for example, left apex was 4+4 cancer. Then, to graphically present a simplified two-dimensional view of the results, the 2-D prostate template shows background prostate in the entire figure (with the assumption that since other 11 cores were negative, we presumed the rest of the prostate to be benign), except for the left apex that shows a representative image of 4+4 cancer placed only in that section of the figure.
Line 77-78: author mentioned a single surgeon did 3600 LRP in 9 years, which is 400 per year. Could I know how long each surgery takes in your hospital?
Reply: The current study demonstrates a single surgeon experience (A.T). The surgeon is considered a figure in RALP over the country, being in a tertiary referral centre made his volume of 400-500 cases a year. The majority of cases are discharged at the morning next to surgery day.
Thanks for your reply. What is catering population in your health centre?
Reply: Approximately 7.5-8 million a year. Our Health centre has 7 branches all over New York City.
Line 91: MRI brand is missing here.
Reply: 3‐T MRI Siemens Skyra equipped with a phased‐ array coil was utilized. This now has been mentioned in the manuscript.
Thanks.
Line 98: internal or external validation needs to be done to verify the MRI data.
Reply: while this is beyond the scope of the current study, we previously published our series to verify MRI data in another separate study specifically designated for this purpose. Please review reference “9”.
Reference 9 only analysed 200 negative MRI patients; this paper also mentioned the limitation of small sample size. Internal and/or external validation is very important for research with large cohort of patients which couldn’t be ignored.
Reply: Thank you for suggestion. Yes, Reference 9 analysed limited number of patients. However, that data was validated in an entirely different cohort [University of Miami]. Also, it is worth to mention regarding Reference 10, we mentioned clinical characteristics of 300 men with non-suspicious MRI that underwent RALP. These men had similar Gleason Grade Groups, positive surgical margins and BCR compared with men having suspicious MRI findings.
Line 105: biopsy methods are not clearly described. See main comment 2.
Reply: Thank you for your suggestions. this was addressed in comment 2
Line 109: the same pathologist reviewed both biopsy and final pathology results which will cause biased results. The biopsy results, MRI results and final pathology results need to be double blinded and then collected and analysed by a third party.
Reply: As per our institutional protocol, the prostate biopsy and prostatectomy specimens are assigned to a specific pathologist with an extensive experience in the field of Prostate cancer. We disagree with the reviewer and we believe that assigning biopsy and final pathology to different pathologists will be intuitively impacted by unavoidable difference in experience and will ultimately lead to bias and heterogeneous results. To support this opinion, we previously published multiple studies that we included our same pathologist in first class urology journals and we did not receive similar comments “please see reference 9” as an example.
Line 120, Gleason Grade Group is mentioned here but there it no reference and definition to show what it is, same for csPAs.
Reply: Thank you for your suggestions. This now has been mentioned in detail with the reference
Line 123-126: the numbers don’t match, which may invalid all analyses.
Reply: We reviewed the number and it matches our study cohorts. Of note, these numbers are lobe specific.
Please refer to main comment 2.
Line 185-186, figure 3, the second and third pathology map are similar? Any description to show how this map was drawn based on 12 biopsy cores?
Reply: the figure was reviewed, and second and third pathology map are totally different as mentioned in the description below each map. Second map shows prostate biopsy finding of GG4 on left side. While third map shows final pathology on radical prostatectomy specimen of the same patient showing GG4 on left side as well as GG4 on right side and additionally positive surgical margin on right side. This scenario clearly demonstrates simplifies our main of the study to better understand “Stealth “contralateral cancers.
Biopsy cores are always discrete and how these cores could be generated to a schematic map similar to the radical prostatectomy one?
See example from figure 2(a) from the following paper to show 12 discrete biopsy cores: https://onlinelibrary.wiley.com/doi/10.1111/iju.13722
Reply: Thank you for your suggestion. Above reference was helpful for methodology section which is added to revised draft. We understand biopsy cores are discrete and can not be generated to schematic map. Also biopsy is equivalent to approx. 0.04% of final specimen. This is mentioned in discussion and limitation section.

Reviewer 3 Report
Revisions:
1, I did not understand the replaced 2 coronal MRI images. This should be very simple correction if authors asked radiologists to help.
2, MRI provided "...minimal benefit..?" still not correct. Many MRI missed lesions did not affect patients' treatment decisions. Why did these missed cases go to surgeries? That is because MRI and TRUS bx already diagnosed significant PCa in these patients. Those missed lesions in other lobes usually did not change patients' outcome.
Author Response
Reviewer 3
1, I did not understand the replaced 2 coronal MRI images. This should be very simple correction if authors asked radiologists to help.
Reply: Thank you for suggestion. Coronal images have been corrected as per suggestion.
2, MRI provided "...minimal benefit..?" still not correct. Many MRI missed lesions did not affect patients' treatment decisions. Why did these missed cases go to surgeries? That is because MRI and TRUS bx already diagnosed significant PCa in these patients. Those missed lesions in other lobes usually did not change patients' outcome.
Reply: Thank you for suggestion. We acknowledge mpMRI followed by targeted and systematic prostate biopsy have revolutionized the scope of prostate biopsy. We removed the statement regarding benefit. Also this cohort is very selective as patients were diagnosed with cancer that needed definitive treatment. We also made this clear in limitations.

Round 3
Reviewer 2 Report
Authors' replies are sufficient to clear my queries, please see my attached file in detials.
